# Macrocycle-stabilization of its interaction with 14-3-3 increases plasma membrane localization and activity of CFTR

Loes M. Stevers [1], Madita Wolter[1], Graeme W. Carlile[2], Dwight Macdonald [3], Luc Richard [3], Frank Gielkens[1], John W. Hanrahan[2], David Y. Thomas[2], Sai Kumar Chakka[3], Mark L. Peterson [3], Helmut Thomas[3], Luc Brunsveld [1] & Christian Ottmann [1]✉

Impaired activity of the chloride channel CFTR is the cause of cystic fibrosis. 14-3-3 proteins have been shown to stabilize CFTR and increase its biogenesis and activity. Here, we report the identification and mechanism of action of a macrocycle stabilizing the 14-3-3/CFTR complex. This molecule rescues plasma membrane localization and chloride transport of F508del-CFTR and works additively with the CFTR pharmacological chaperone corrector lumacaftor (VX-809) and the triple combination Trikafta®. This macrocycle is a useful tool to study the CFTR/14-3-3 interaction and the potential of molecular glues in cystic fibrosis therapeutics.

[1] Laboratory of Chemical Biology, Department of Biomedical Engineering and Institute for Complex Molecular Systems, Eindhoven University of Technology, Eindhoven, The Netherlands. [2] Cystic Fibrosis Translational Research Centre, Department of Biochemistry and Physiology, McGill University, Montreal, QC, Canada. [3] Cyclenium Pharma Inc., 7171 rue Frederick Banting, Montreal, Quebec, Canada. ✉email: c.ottmann@tue.nl

The CFTR protein is a cyclic adenosine 5′-monophosphate (cAMP)–regulated transporter with anion channel activity that conducts Cl⁻ on the apical surface of bronchial epithelial cells. The F508del-CFTR mutation is the most frequent cause of cystic fibrosis (CF)[1]. It codes for a mutant protein that is recognized as misfolded and retained in the endoplasmic reticulum but, if induced to traffic to the plasma membrane, it is almost fully functional[2]. Cell-based assays have identified compounds that facilitate the trafficking of F508del-CFTR to the plasma membrane. So far, the most effective compounds for F508del-CFTR have been pharmacological chaperones that bind to the mutant CFTR molecule and assist its correct folding[1]. To amplify the function of the corrected F508del-CFTR, the potentiator ivacaftor (Kalydeco®) that corrects the G551D mutation (which traffics normally but has defective gating) is included in combination with pharmacological chaperones[3]. There have been two approvals of such drug combinations that correct the trafficking of the F508del-CFTR, ivacaftor/lumacaftor (Orkambi®)[4] in 2015 and ivacaftor/tezacaftor (Symdeko®)[5] in 2018. More recently (2019), the triple combination ivacaftor/tezacaftor/elexacaftor (Trikafta®) has been approved[6]. The need for these combinations is that single molecules give low levels of correction and, although a new corrector VX-445 (elexacaftor) has been recently described that appears to give clinically significant levels of correction over a range of CFTR mutations, optimal results still require combination therapies[7,8].

These molecules are targeting CFTR directly and function as molecular chaperones. In particular, ivacaftor and an investigational drug from Galapagos (GLPG1837) have been shown to bind CFTR at the protein/plasma membrane interface with half of the molecules' surfaces exposed to the lipid bilayer[9]. Although the above-described drugs represent breakthrough therapies for cystic fibrosis, it is still highly informative to explore further aspects of CFTR biology for modulation by small molecules to potentially improve clinical outcomes. For example, protein-protein interactions (PPIs) have been the focus of drug discovery and chemical biology for some time[10–13], triggered by the seminal examples of the natural products rapamycin and FK506[14], the approval of venetoclax (Venclexta®) as the first Bcl-2 inhibitor[15], and the tremendous clinical and economic success of lenalidomide (Revlimid®)[16]. In 2015, Pankow et al. reported a CFTR interactome analysis of both wildtype (wt) and F508del-CFTR[17]. In this study, 638 partner proteins of CFTR were identified and an extensive remodeling of the F508-del interactome upon rescue of mutated CFTR function by low temperature (26–30 °C) or HDAC inhibition was demonstrated. Furthermore, RNA interference was used to identify proteins whose specific knockdowns rescue or reduce CFTR function in the F508-del mutant, many of which are involved in the degradation machinery, quality control, or membrane trafficking processes[17]. These findings suggest that modulation of the CFTR interactome could contribute to a therapeutically beneficial restoration of impaired CFTR function. Interestingly, among the proteins that in this study were consistently found in the CFTR interactome of both wt and F508-del, were the 14-3-3 proteins, important regulators of Ser/Thr-phosphorylated proteins.

14-3-3 proteins are dimeric proteins that have been shown to bind to the disordered regulatory (R) domain of CFTR, facilitate trafficking to the plasma membrane and enhance ion channel activity[18,19]. We have previously reported the crystal structure of 14-3-3 in complex with a number of phosphopeptides derived from CFTR and have shown that the natural product fusicoccin A (FC-A) can stabilize the 14-3-3/CFTR interaction and promote plasma membrane localization of F508del-CFTR[20]. Since the structural complexity of fusicoccanes poses a significant challenge for medicinal chemistry optimization, we sought to identify other synthetic chemotypes that are able to stabilize the 14-3-3/CFTR complex.

In this work, we report the identification, structural characterization and cellular activity of a class of macrocycles stabilizing the 14-3-3/CFTR PPI.

## Results and discussion

**Identification of macrocycle-stabilizers of the 14-3-3/CFTR complex.** We screened 5760 compounds of Cyclenium's proprietary small-molecule macrocycle library employing a fluorescence polarization (FP) assay for the binding of the di-phosphorylated CFTR-derived synthetic peptide CFTRpS753pS768 to 14-3-3β (Fig. 1a, b, Supplementary Table 1). In the single concentration screen, 24 hits were identified (Fig. 1c) of which seven (7) showed concentration-dependent stabilization (Fig. 1d, e). These compounds can be grouped into four (4) chemotypes (Fig. 1f) which prompted us to design and synthesize a second, focused validation library of 480 macrocycles. FP screening of this library added another eight (8) compounds (Fig. 2a) as validated stabilizers of the 14-3-3β/CFTRpS753pS768 interaction to the initial set (Fig. 2b). In the presence of these compounds, the apparent $K_d$ for the CFTR peptide binding to 14-3-3 was increased significantly (Supplementary Fig. 1).

**The co-crystal structure of the 14-3-3/CFTRpS753pS768/ CY007424 complex reveals a different binding mode and explains its additive effect with FC-A.** In order to gain structural information on the PPI stabilizing activity of these compounds, co-crystallization trials of 14-3-3β in complex with eight (8) of the validated hits were performed. Diffraction-quality crystals could be optimized for the complex with CY007424 (Fig. 3a) and the structure was solved at 1.76 Å. The 14-3-3β/CFTRpS753pS768/ CY007424 complex crystallized as a tetramer in the asymmetric unit with two 14-3-3 dimers, two copies of the CFTR peptide and two molecules of CY007424 (Supplementary Fig. 3) with electron density covering the entire macrocyclic molecule (Fig. 3b, c). Of the 28 residues of the CFTRpS753pS768 peptide, 21 amino acids could be built into the model (Fig. 3d). CY007424 binds close to pS753 of the CFTR peptide and establishes contacts to 14-3-3, as well as to the peptide (Fig. 3c). The tyrosine moiety of CY007424 is embedded in a shallow hydrophobic cleft formed by Pro750 and Ile752 of CFTR and Leu229 of 14-3-3β (Figs. 3c, d and 4a). The two main ring thioether phenyls are engaged in a hydrophobic interaction with the hydrocarbon part of Arg58, Ser59 and Arg62 of 14-3-3 (Figs. 3c, d and 4b). There is a polar interaction between the main chain nitrogen and carbonyl oxygen of Arg751 of CFTR and the corresponding ring nitrogen and carbonyl of the tyrosine moiety of CY007424 (Fig. 4c). A third polar contact is established between the arginine moiety of CY007424 and Asn52 of 14-3-3β (Fig. 4d).

Comparison of the binding mode of the previously established FC-stabilized CFTRpS753 interaction motif with the current CY007424-stabilized structure revealed a considerable conformational change induced by CY007424 in the N-terminus of the peptide (Fig. 3e). The positions of P750 and R751 flip by around 180° and the side chains of L749 and R751 become visible in the electron density. In the case of R751, this allows a polar interaction between the phosphate of pS753 and the terminal amino group of this arginine, as well as a direct contact with CY007424 (Fig. 4d). The observed conformational changes are necessary to establish the above-described binding mode, especially the accommodation of the tyrosine moiety of CY007424. This means that CY007424 either "selected" the observed peptide conformation from an ensemble of states that

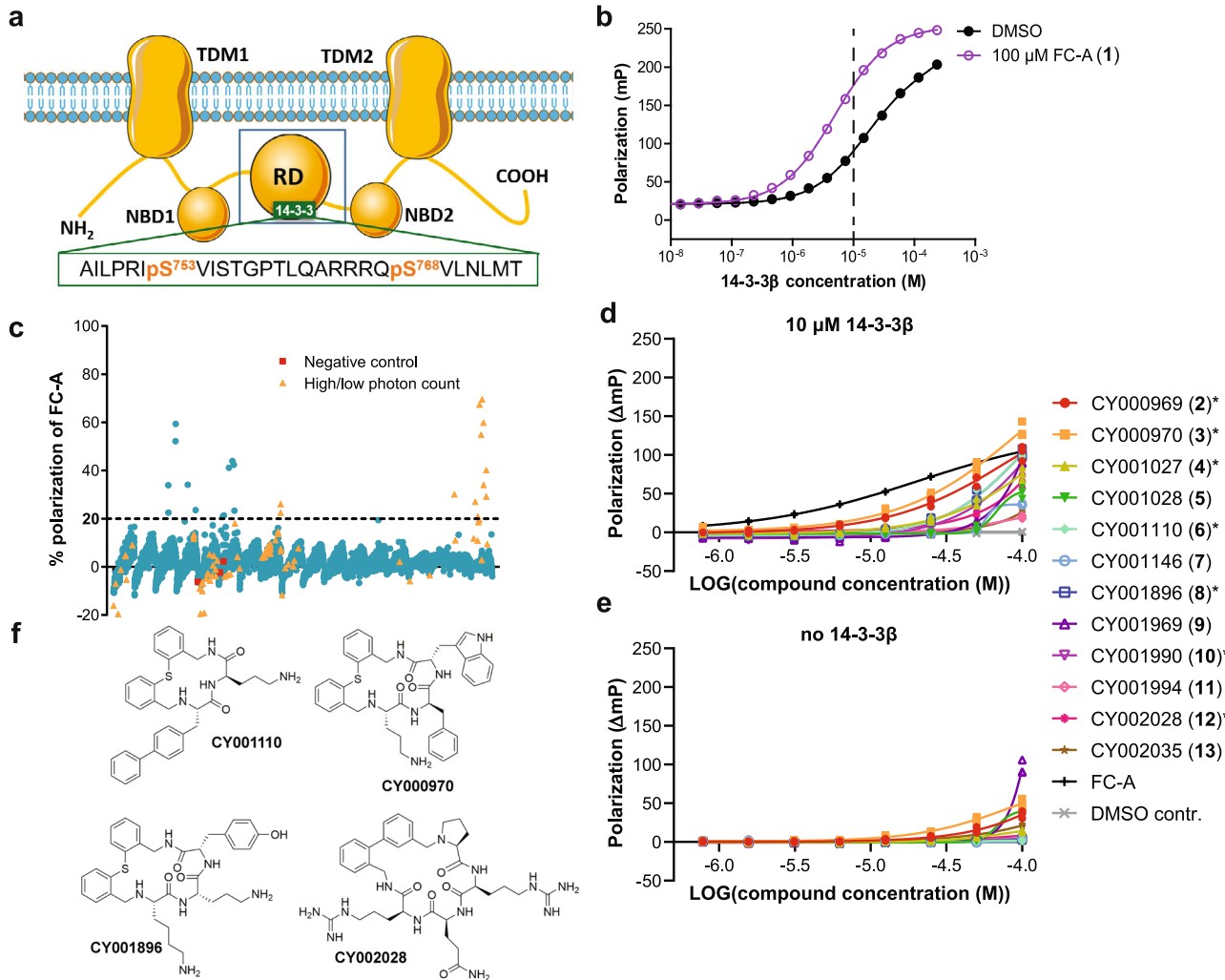

**Fig. 1 Screening for stabilizers of the 14-3-3β/CFTRpS753pS768 complex. a** Topology model of CFTR showing the two transmembrane domains (TDM), the nucleotide binding domains (NBD), and the regulatory domain (RD). The 14-3-3 binding motif encompassing the phosphorylation sites pS753 and pS768 (CFTRpS753pS768) was used in this study and is shown as peptide sequence. This sequence—CFTRpS753pS768—was used in this study as a synthetic peptide. **b** Fluorescence polarization (FP) assay of fluorescein isothiocyanate (FITC)-labeled CFTRpS753pS768 peptide (100 nM) with 14-3-3β, in the absence (black) or presence (purple) of 100 μM fusicoccin A (FC-A), $n = 3$ technical replicates. The 14-3-3β concentration used for the FP screening assay was 10 μM (dashed line). **c** HTS FP assay of the Cyclenium library. The samples contain 100 nM of FITC-CFTR_pS753pS768 peptide, 10 μM 14-3-3β, and 50 μM compound. FC-A (100 μM) was added as a positive control and DMSO as the negative control. Hit compounds are defined as a polarization increase which is higher than 20% of the response of FC-A. Red squares are compounds that appeared positive in the negative control without 14-3-3. Orange triangles gave a suspicious high or low total photon count and need to be treated with care. **d** The eight-point dose-response FP follow-up assay of the Cyclenium library hit compounds stabilizing the interaction between 14-3-3 (10 μM) and labeled CFTRpS753pS768 (100 nM) peptide. Background polarization was subtracted from all values, $n = 3$ technical replicates. Compounds selected for further analysis are indicated with an asterisk. **e** Polarization in the absence of 14-3-3. Background polarization was subtracted from all values, $n = 3$ technical replicates. **f** Representatives from the four chemotypes of active compounds after screening for stabilizers of the 14-3-3β/CFTRpS753pS768 complex. Source data are provided as a Source Data file.

this flexible peptide can adopt or an "induced-fit" adaptation of the peptide took place in the presence of the macrocycle.

In accordance with the different binding sites of the two 14-3-3/CFTR stabilizers FC-A and CY007424, simultaneous treatment resulted in an additive effect and increased the apparent affinity of the CFTR peptide to 14-3-3 by almost three orders of magnitude (x 809), from 17 μM to 21 nM (Fig. 3f). CY007424 is a much more efficacious compound than FC-A[20], stabilizing the complex by a factor of more than 300x, whereas FC-A shows an about 4.5x stabilization at a concentration of 100 μM (Fig. 3f). At very high 14-3-3 concentrations, low-affinity background binding of CY007424 to 14-3-3apo most probably lowers the amount of compound available for stabilization of CFTR peptide binding.

Pull-down experiments reveal the stabilizing effect of CY007424 on the 14-3-3/CFTR interaction in a cellular context (Fig. 3g). In comparison, pull-down experiments with CY007491 and CY007476 did not show a stabilizing effect of the 14-3-3/full-length CFTR interaction in cells (Supplementary Fig. 4), a result that could explain the missing stimulatory effect on CFTR ion conductance (see below).

In order to test the specificity of CY007424 in stabilizing the 14-3-3/CFTRpS753pS768 complex, we measured binding of a number of 14-3-3 partner protein peptides (from BAD, Foxo3, Foxo4, IRS1, MDM2, p53, RND3, and RPTOR) in the absence and presence of the compound. With the peptides derived from BAD and IRS1, we could observe some stabilization of the

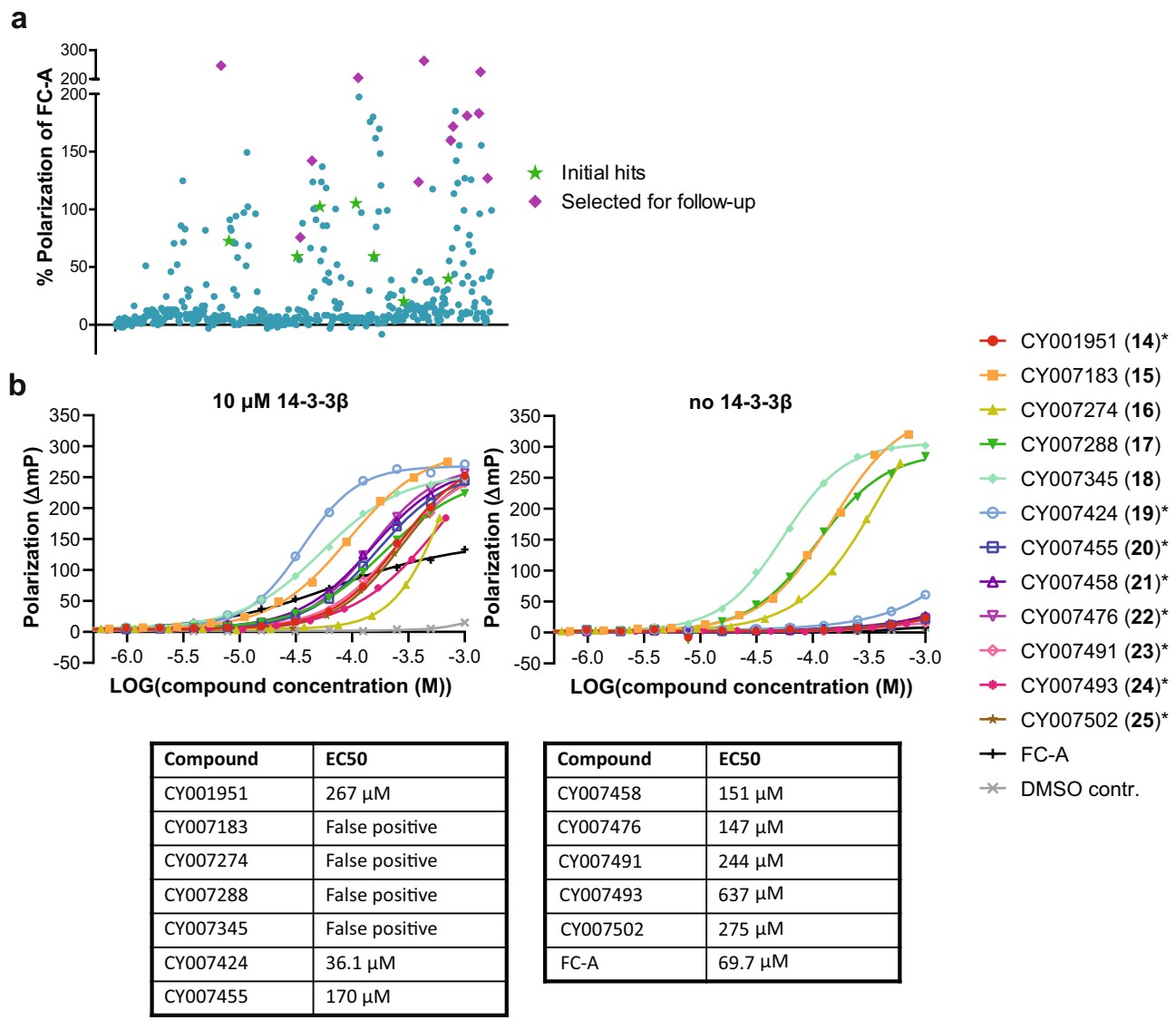

**Fig. 2 Screening results of the hit validation library that consisted of resynthesized primary hit compounds and newly designed macrocycles to explore structure-activity relationships. a** HTS FP of all 480 compounds. The samples contain 100 nM of FITC-CFTRpS753pS768 peptide, 10 μM 14-3-3β, and approximately 125 μM compound, dependent on the compound stock concentration. The positive control was FC-A (100 μM) and negative control was DMSO. Green stars are the initial hit compounds from previous library and the purple diamonds are selected for the follow-up dose-response assay. **b** Dose–response FP follow-up assay of the selected hit compounds stabilizing the interaction between 14-3-3β (10 μM) and labeled CFTRpS753pS768 (100 nM) peptide. Background polarization was subtracted from all values, $n = 3$ technical replicates. Compounds selected for further analysis are indicated with an asterisk. Source data are provided as a Source Data file.

**CY007424 increases plasma membrane localization and ion transport of F508del CFTR and works additively with VX-809.** Twelve (12) validated hit compounds from the initial screening and focused validation library were tested for their effect on F508del-CFTR trafficking. Baby hamster kidney (BHK) cells expressing 3HA-tagged F508del-CFTR were treated for 24 h with 10 μM compound. After fixation of the cells, a combination of mouse monoclonal anti-HA antibody and anti-mouse IgG conjugated with FITC was used to detect F508del-CFTR that had trafficked to the plasma membrane as described previously[21]. CY007424 showed the strongest increase of F508del-CFTR trafficking towards the plasma

complex (Supplementary Fig. 2) in the presence of 100 μM CY007424, however this effect was profoundly weaker than with the CFTR peptide (Fig. 3f).

membrane, followed by CY007491, and CY007476 (Fig. 5a). Additionally, a fluorescence imaging plate reader (FLIPR) Membrane Potential (FMP) assay was performed. This assay measures real-time membrane potential changes associated with ion channel activation and ion transporter proteins. F508del-CFTR expressing BHK cells were incubated for 24 h with 20 μM compound before treatment with the potentiator genistein, FMP dye, and activator forskolin. The fluorescence intensity is thus a measure for the function of the CFTR protein in the plasma membrane of these cells. The F508del-CFTR corrector VX-809 (lumacaftor) and the triple combination Trikafta® were used as positive controls in this assay. CY007424 has a clear corrector function on F508del-CFTR, while the other macrocycles show no significant increase in CFTR function compared to the DMSO control (Fig. 5b). This is in line with the observed effects in a pull-down experiment with 508Fdel-CFTR expressing HEK293 cells, where CY007424 increased CFTR

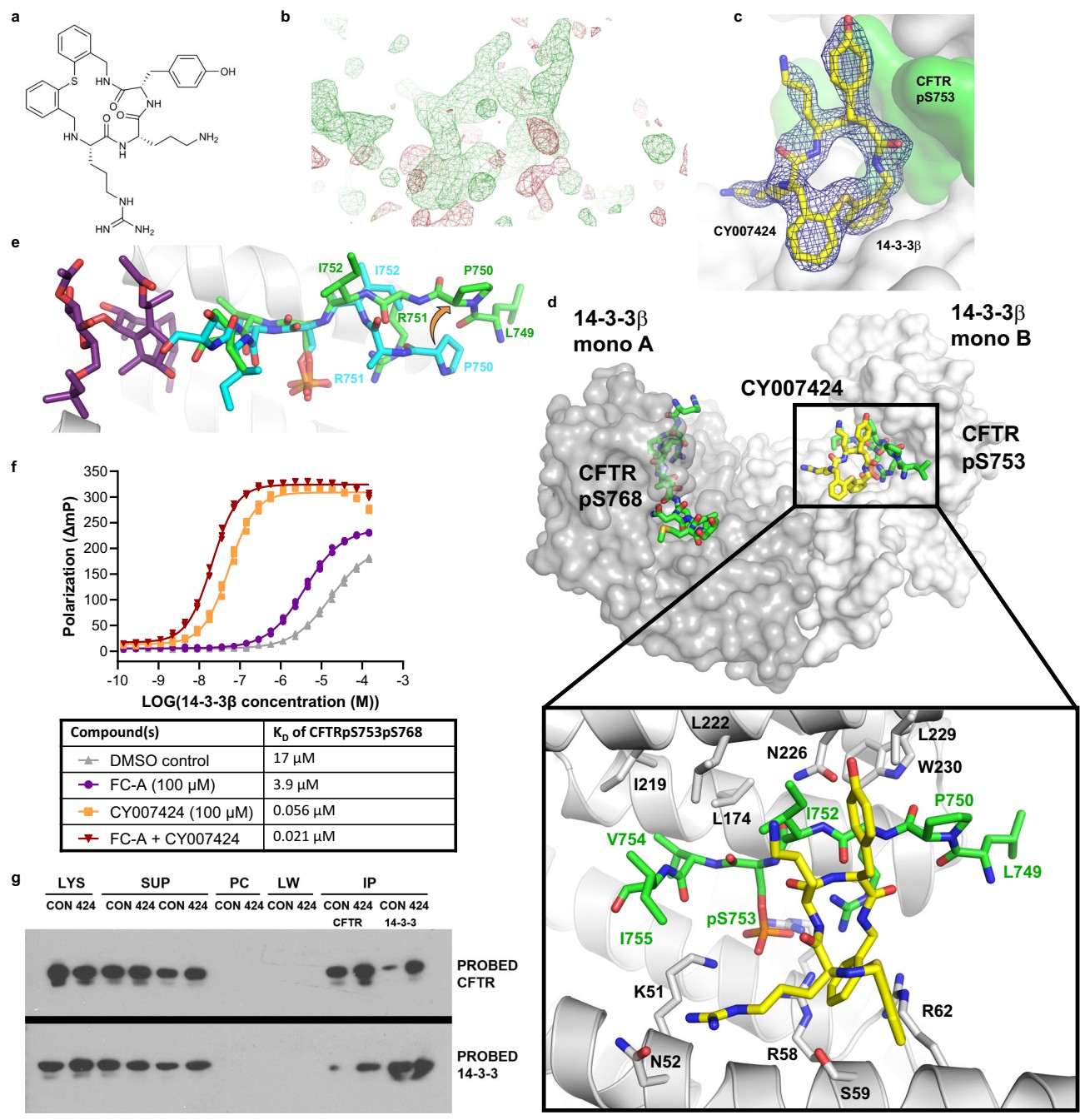

**Fig. 3 Macrocycle stabilization of the 14-3-3/CFTR interaction. a** Chemical structure of CY0072424. **b** Electron density map (unbiased Fo-Fc, 2.5σ, directly after molecular replacement) of CY007424 bound to the interface of 14-3-3β and CFTRpS753. **c** Final 2Fo-Fc electron density map (1σ). **d** Crystal structure of the 14-3-3β homodimer complexed with CFTRpS753pS768 and CY007424 (PDB ID: 7QI1). **e** Overlay of the CFTRpS753 peptide motif bound to 14-3-3 in the presence of FC-A (cyan sticks) or CY007424 (green sticks). FC-A is shown as purple sticks. **f** FP assay of FITC-labeled CFTRpS753pS768 (10 nM) with 14-3-3β in the presence of 100 μM of FC-A, CY007424, or both compounds. Background polarization was subtracted from all values, $n = 3$ technical replicates. **g** Immunoprecipitation study to show the effect of CY007424 on the interaction between of CFTR and 14-3-3 proteins. CFTR and separately 14-3-3 immunoprecipitation from HEK293 cells expressing F508del-CFTR after 24 h of CY007424 (10μM) (424) or vehicle alone (CON). LYS: cellular lysate prior to preclear, SUP: supernatant after specific bead incubation, PC: second preclear, LW: last wash after the immunoprecipitation IP are the specific antibody beads. IP, upper blot: probed with CFTR AB, lower blot: probed with 14-3-3 AB. Source data are provided as a Source Data file.

binding to 14-3-3 (Fig. 3g), while CY007491 and CY007476—which are inactive in the ion transport assay—did not enhance binding of 14-3-3 during the co-immunopreciptitation (Supplementary Fig. 4). When the cells were incubated with the CFTR inhibitor 172 (INH172) the effects in this assay were neutralized, demonstrating CFTR-dependency of the signal increases (Fig. 5c). Interestingly, the combinations of VX-809 and CY007424 and Trikafta® and

CY007424 show an additive effect on CFTR function in the cell membrane (Fig. 5c). Another cellular assay was performed with the Ussing Chamber, which detects and quantifies transport of ions across epithelial tissue[22]. F508del-CFTR expressing cystic fibrosis epithelial (CFBE) cells were treated for 18 h with 20 μM CY007424. After forskolin and genistein stimulation, the short-circuit current of the cells was measured. The CY007424-treated CFBE cells

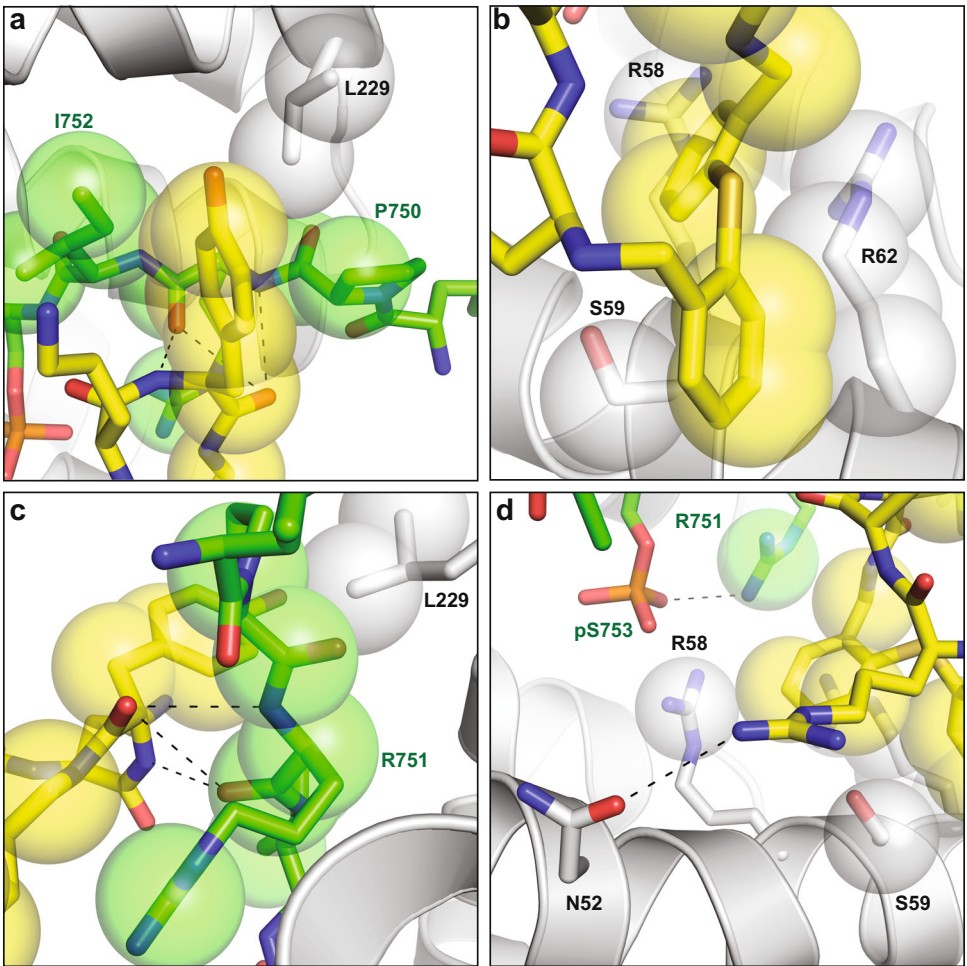

**Fig. 4 Detailed view of the interactions of CY007424 binding to the complex of 14-3-3β and CFTRpS753pS768. a** Interaction of the tyrosine moiety of CY007424 (yellow sticks and semi-transparent spheres) with P750 to I752 of the CFTRpS753pS768 peptide and L229 of 14-3-3β. **b** Binding of the thio-bis-phenyl part of the core ring of CY007424 to the hydrocarbon part of R58, S59, and R62 of 14-3-3β. **c** Polar interactions of the main-chain nitrogens and carbonyls of the tyrosine moiety of CY007424 and R751 of CFTRpS753pS768. **d** Polar contact of the guanidinium group of CY007424 with N52 of 14-3-3β. In the presence of CY007424, the sidechain of R751 of CTRFpS753pS768 becomes visible and establishes a polar contact with the phosphate of pS753.

showed a higher conductance than the DMSO control treated cells. Also here, the combination of CY007424 and VX-809 showed an additive response compared to VX-809 alone (Fig. 5d). However, the additive effect of combining CY007424 and Trikafta® seen in the FLIPR assay (Fig. 5c) was not observed in the experiment with CFBE cells in the Ussing chamber (Fig. 5d), highlighting the need for further studies to identify possible useful combinations of existing therapeutic agents with macrocycles that stabilize the regulatory 14-3-3/CFTR complex.

In conclusion, this study demonstrates that synthetic compounds are able to stabilize the interaction of the CF-related chloride channel CFTR with 14-3-3 proteins and reveals the mode of action of the discovered macrocycles. Since 14-3-3 proteins are positive regulators of CFTR that facilitate forward trafficking to the plasma membrane and stabilize the functional fold of the channel, these compounds are useful tools to study the CFTR/14-3-3 interaction.

## Methods

**Reagents and peptides**. Fusicoccin A (FC-A) was obtained from Enzo Life Sciences BVBA. All small molecule macrocycle compounds and libraries were synthesized as described in the Supplementary Methods and provided by Cyclenium Pharma. The CFTR peptides were synthesized as previously described[20].

**Expression of 14-3-3**. His$_6$-tagged 14-3-3 isoforms (full-length and ΔC) were expressed in NiCo21(DE3) competent cells (0.4 mM IPTG, overnight at 18 °C), with a pPROEX HTb plasmid, and purified with a nickel column. The His$_6$-tag was cleaved-off with TEV-protease and a second purification was done by size exclusion chromatography. The proteins were dialyzed against FP, ICT, or crystallization buffers before usage (described below).

**Fluorescence polarization (FP) assay**. The FITC-labeled peptides were dissolved in FP buffer (10 mM HEPES pH 7.4, 150 mM NaCl, 0.1% Tween20, 1 mg/mL BSA) to a final concentration of 100 nM or 10 nM. For the dose-response assays, a 14-3-3β concentration of 10 μM was used while titrating the compounds in a two-times dilution series and, for the other assays, a dilution series of 14-3-3β was made with constant concentration of compound. The dilution series were made in 384 Corning Black Round Bottom well plates and their polarization was measured with a Tecan Infinite F500 plate reader (ex. = 485 nm, em. = 535 nm). The method of fitting of the curves was: "log(agonist) vs. response – Variable slope (four parameters)" from GraphPad Prism. The equation is $Y = Bottom + (Top-Bottom)/(1 + 10^{\wedge}((LogEC50-X)*HillSlope))$.

**HTS FP assay**. The HTS FP assay was set up in 384 Corning Black Round Bottom well plates. Each well contained 20 μL sample with 10 μM 14-3-3β and 100 nM FITC-labeled CFTRpS753pS768 peptide dissolved in FP buffer (10 mM HEPES pH 7.4, 150 mM NaCl, 0.1% Tween20, 1 mg/mL BSA). Compound was added using a pin tool system adding approximately 0.1 μL to each well, dependent on the viscosity of the solution. Each plate contained 16 wells with negative controls (DMSO) and 16 wells with positive control (FC-A). The plates were incubated for 30 min in the dark at RT before measuring the polarization using a PHERAstar FS plate reader (ex. = 485 nm, em. = 520 nm).

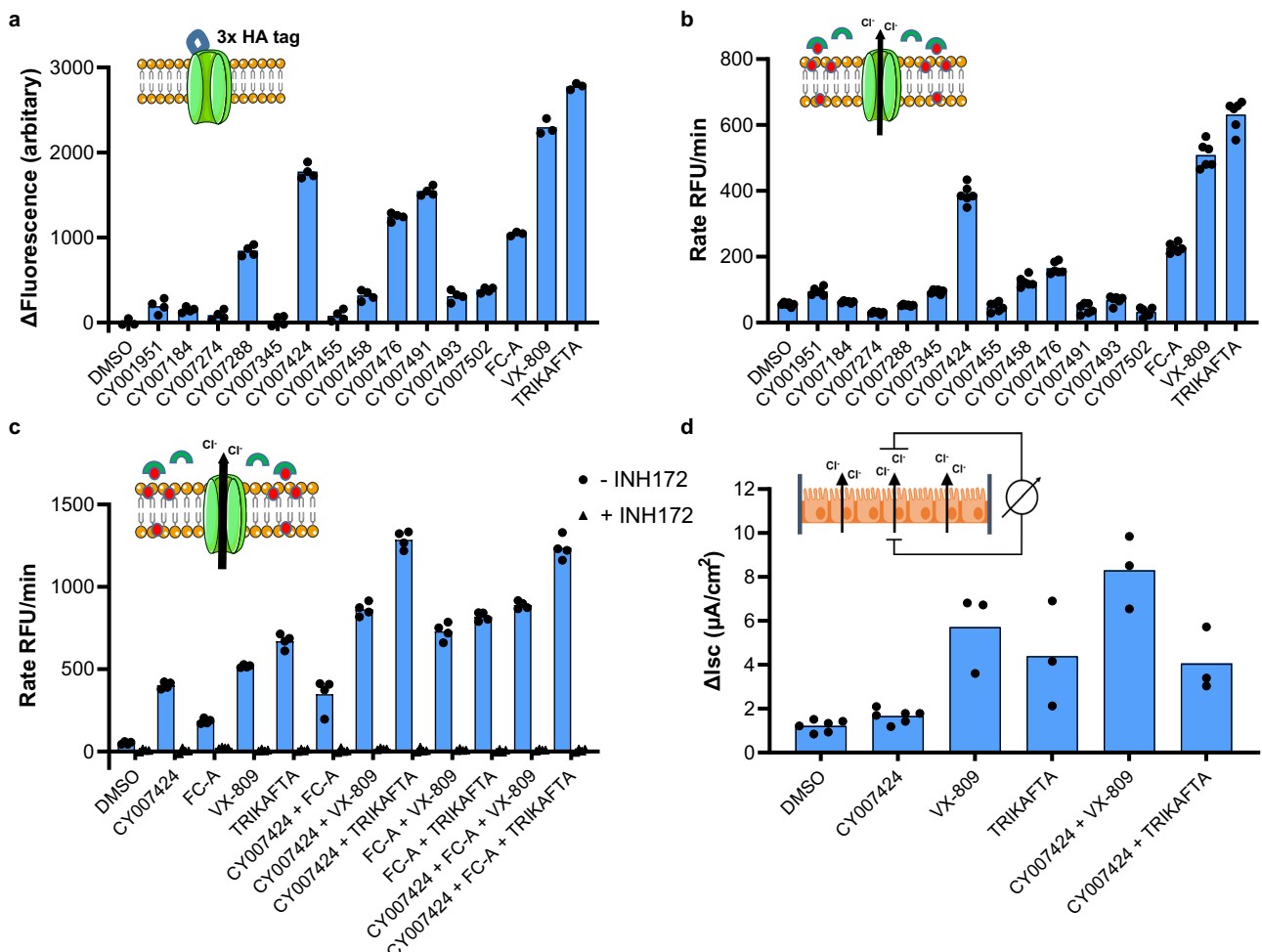

**Fig. 5 Cellular activity of macrocycles stabilizing the 14-3-3/CFTR PPI. a** Surface expression of 3HA-tagged F508del-CFTR in BHK cells incubated for 24 h with 10 μM compound and treated with mouse monoclonal anti-HA antibody and anti-mouse IgG conjugated with FITC ($n = 4$ or $n = 3$ biological replicates, bars represent the mean). **b, c** FLIPR Membrane Potential (FMP) assay of F508del-CFTR expressing BHK cells treated for 24 h with the macrocycles (**b**, $n = 6$ biological replicates, bars represent the mean) or with CY007424 and/or VX-809 or Trikafta®, with and without CFTR inhibitor INH172 (**c**, $n = 4$ biological replicates, bars represent the mean). **d** Using chamber experiment of F508del-CFTR expressing CFBE cells treated for 18 h with CY007424, VX-809, or Trikafta® or in combinations. CFTR activity is assayed by measurement of short-circuit current of cells after stimulation by forskolin and the potentiator genistein ($n = 3$ or $n = 6$ biological replicates, bars represent the mean). Source data are provided as a Source Data file.

**Crystallography**. The 14-3-3β protein was C-terminally truncated after T232 to improve crystallization. For crystallization, the 14-3-3βΔC/CFTR_pS753pS768/CY07424 complex was mixed in a 2:1.5:4 molar stoichiometry with a final protein concentration of 15 mg/mL in crystallization buffer (25 mM HEPES, 0.1 M NaCl, 2 mM DTT, pH 7.4). This was set up for hanging-drop crystallization in a 1:1 ratio with Qiagen Cryos Suite #44 crystallization liquor (0.09 M HEPES sodium salt, pH 7.5, 1.26 M tri-Sodium citrate, 10% (v/v) Glycerol) with an extra 2% of Glycerol (SigmaAldrich). Crystals were fished out after 2 weeks of incubation at 4 °C and flash-cooled in liquid nitrogen. Diffraction data was collected at the PETRA III P11 beamline (DESY, Hamburg, Germany). The dataset was indexed and integrated using XDS[23] and scaled using SCALA[24]. The structure was phased by molecular replacement, using PDB ID 2C23[25] as search model, in Phaser[26], Coot[27] and phenix.refine[28] were used in alternating cycles of model building and refinement. 5% of randomly chose reflections were set aside for the test set. See Supplementary Table 2 for data collection, structure determination, and refinement.

**Immunoprecipitation**. CFTR monoclonal-mAb (M3A7, ab270238 abcam) and 14-3-3 monoclonal-Rbt-Ab (Y62, ab32560 abcam) were separately covalently linked to Protein A–Sepharose (Pharmacia, NJ, U.S.A.) by cross-linking with dimethyl pimelimidate (DMP) utilizing standard procedures.

Briefly, Protein A–Sepharose beads were washed two times in 0.2 M sodium borate, pH 9.0 for 5 min each. The beads were then mixed with antibody (for 1 mL of beads use 0.5 mg of antibody). These were placed on a rotator and mixed continuously for 2 h at room temperature (RT). The beads were washed with sodium borate twice, followed by one wash with 0.2 M triethanolamine, pH

8.5. An equal volume of 40 mM DMP in 0.2 M triethanolamine was added. The mixture was mixed for 1 h at RT and the beads then collected. The beads were incubated in 0.2 M ethanolamine, pH 8.2 for 5 min at RT. Beads were washed with the sodium borate solution, and stored at 4 °C until use. Beads were rinsed in lysis buffer before use.

Once HEK 293 cells expressing F508del-CFTR (cells earlier described by Carlile et al. 2015)[29] were treated, they were harvested and immunoprecipitated as follows. Cells were washed twice in sterile TBS, trypsin-treated and transferred to a Falcon tube with media and sera and centrifuged at 800 g at RT for 5 min. Pellets were washed twice in TBS prewarmed to 37 °C and centrifuged at 800–1000 g for 5 min at 4 °C. Cells were lysed in buffer IPB [150 mM NaCl, 20 mM Tris-HCl (pH 7.4), 1% Nonidet P40, 100 μM PMSF and 5 μg/ml of the following protease inhibitors: chemostatin, leupeptin, pepstatin A, aprotinin and antipain] on ice for 20 min. The list was centrifuged at 13,000 g for 20 min at 4 °C. Supernatants were collected, and precleared twice using 20 μl of Protein A–Sepharose covalently linked to mouse IgG (or rabbit IgG) for at least 1 h at 4 °C. Samples were centrifuged in order to remove beads (800 g for 5 min at 4 °C). Protein A–CFTR or Protein-A-14-3-3 beads were added to the supernatant and incubated with mixing for 2 h at 4 °C. The supernatant was removed and collected. Beads were then washed three times with IPB, and an additional three times with modified IPB [containing 0.1% (v/v) deoxycholate and no Nonidet P40]. Beads were centrifuged and collected. An equal volume of reducing sample buffer was added in preparation for SDS PAGE.

**CFTR trafficking assay**. The trafficking assay was performed as previously described by Carlile et al.[21]. In brief, 3HA-tagged F508del-CFTR expressing baby hamster kidney

(BHK) cells (cells earlier described by Carlile et. al. 2007)[21] were seeded in 96-well plates (Corning half area, black-sided, clear bottom) at 15,000 cells per well. After 24 h incubation at 37 °C, the cells were treated with 10 μM of compound for 24 h (final DMSO concentration 1% v/v). The cells were fixated with 4% paraformaldehyde, washed with PBS, and then blocked with fetal bovine serum (5% in PBS). Mouse monoclonal anti-HA antibody (H9658 Sigma, 1:150 dilution in PBS) was incubated overnight, and, after three washes with PBS, the background fluorescence was measured in a plate reader (488 nm excitation, 510 nm emission). The secondary antibody anti-mouse IgG conjugated with FITC (F5262 Sigma, 1:100 dilution in PBS) was incubated for 1 h, the cells were washed three times with PBS and analyzed in the plate reader again. Background fluorescence was subtracted from the signal, after which the signal was normalized to the DMSO control and wt-CFTR expressing cells.

**FLIPR membrane potential assay**. The FLuorescence Imaging Plate Reader (FLIPR) Membrane Potential (FMP) assay is based on the technique developed by Van Goor et al.[30] F508del-CFTR expressing baby hamster kidney (BHK) cells were incubated with the compounds for 24 h at 37 °C. The growth medium was removed from the cells by inverting the plate and FMP dye (Molecular Devices, Part #R8042) including the potentiator genistein (Sigma, G6649) was added back in 70 μl of low Cl⁻ containing buffer (160 mM NaGluconate, 4.5 mM KCl, 2 mM CaCl$_2$, 1 mM MgCl$_2$, 10 mM D-Glucose, 10 mM HEPES (pH7.4)). The plates were incubated for 5 min at RT before activation of CFTR in the plate reader (SynergyMX) with the addition of 14 μL of FMP dye in low Cl⁻ buffer containing 6x forskolin (Sigma, F6886) following a 2 min baseline read. Fluorescence intensity was monitored for 5 min following CFTR activation. Reported is the rate of fluorescence intensity change over time.

**Ussing chamber assay**. Primary F508del-CFTR expressing CFBE cells (cells earlier described in Carlile et al. 2016)[31] were grown in air-liquid interface culture conditions until fully differentiated. The cells were treated for 18 h with 20 μM CY07424, 1 μM VX-809 or the combination of the two compounds. Once the cells were mounted in the Ussing Chambers, 10 μM of forskolin, the potentiator genistein and CFTR inhibitor INH172 were added in the chambers before measuring the short-circuit current over the CFBE cells as described by Hug et al.[32].

**Reporting summary**. Further information on research design is available in the Nature Research Reporting Summary linked to this article.

## Data availability

Source data are provided with this paper. The crystal structure solved in this study is available in the Protein Data Bank (PDB) as PDB entry 7QI1 [https://doi.org/10.2210/pdb7QI1/pdb]. The previously published structure used here is available as PDB entry 2C23 [https://doi.org/10.2210/pdb2C23/pdb]. All other data are available from the corresponding author on request. Source data are provided with this paper.

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

## Acknowledgements

J.W.H., G.W.C., and D.Y.T. are supported by grants from the Canadian Institutes of Health Research and Cystic Fibrosis Canada. D.Y.T. is the Canada Research Chair in Molecular Genetics. This work is funded by the H2020 Marie Curie Actions of the European Commission through the TASPPI project, grant Agreement 675179 and by the Netherlands Organization for Scientific Research via Gravity Program 024.001.035.

## Author contributions

L.M.S. and M.W. performed biochemical and crystallographic experiments, G.W.C. performed cell experiments, D.M., L.R., and F.G. synthesized macrocycles, J.W.H., D.Y.T., S.K.C., M.L.P., H.T., L.B., C.O. analyzed data. All authors contributed to writing the manuscript.

## Competing interests

L.B. and C.O. are founders and shareholders of Ambagon Therapeutics, C.O. and L.M.S. are employees of Ambagon Therapeutics. The remaining authors declare no competing interests.
