## [Peer Review File · Nature Communications]

REVIEWER COMMENTS

Reviewer #1 (Remarks to the Author):

In this manuscript, Stevers et al. describe the discovery of macrocyclic compound that upregulates the interaction of 14-3-3 and a phosphopeptide fragment of CFTR. The crystal structural analysis of 14-3-3 β bound to the phosphopeptide CFTRpS753pS768 and CY0072424 reveals the new mechanism of stabilizing 14-3-3-ligand interaction upon the formation of ternary complex: The compound binds to distinct part of 14-3-3 and the peptide than fusicoccin-A (FC-A) does, resulting in the remarkable stabilizing effect. The results of biological evaluation using BHK cells expressing 3HA-tagged F508del-CFTR show that CY007458 resulted in the highest signal of membrane-recruited CFTR molecules and their channel activity, suggesting that the compound promotes translocation of CFTR to plasma membrane.

The presented results overall are very interesting. Particularly, the structural analysis study based on the X-ray crystallography have clearly revealed the new mechanism of action of the macrocycle for stabilizing 14-3-3 interaction. This results in the significant enhancement of the peptide binding by 3-ordered magnitude compared to peptide alone, which is very impressive. Intriguingly, the synergetic effect of CY0072427 and VX-809 on the efflux of chloride ion suggests that translocation of mutated CFTR is promoted by the enhancement of 14-3-3-binding in the presence of CY0072427, and that proper folding of the resulting translocated CFTR is assisted by VX-809, leading to better biological outcome compared to VX-0-809 alone. This result suggests the potential clinical application of combination therapy of 14-3-3-enhancer and the conventional CFTR-correctors for cystic fibrosis. Positive modulation of protein-protein interactions still remains challenging task. In my opinion, this paper is sufficiently interest to be accepted after some points have been resolved or corrected.

1) In Fig. 1: Please add the sequential information of the fluorescently labeled peptide used in the screening (S.I.) and the FP assay (Fig. 1e). According to the description "CFTRpS753pS768", the peptide seems to be doubly phosphorylated, whereas the peptide shown in the crystal structure (green sticks) is mono phosphorylated. Please clarify the difference.

2) In Fig. S4: Please add the apparent K_d values of each compound. In addition, I wonder why CY007424 presents a bell-shaped curve. This may suggest that the compound may form assembly at the low concentration of 14-3-3 and off from 1:1 binding. Please explain.

3) Does the mutation of F508 affect 14-3-3-binding of CFTR? Would it be possible that CY007424 is selective for mutated CFTR over normal CFTR?

4) It might be interesting to see if CY007424 is selective for the CFTR peptide. This is likely the case as the crystal structure reveals the tyrosine residue of the compound makes significant hydrophobic contact with the peptide. I suggest that the authors test several peptides for binding to 14-3-3 in the presence of CY007424.

5) In Fig. 2a and b: Please discuss the structure-activity-relationship for the biological activities. There is inconsistency between the K_d values and the cell-based activities presented in Fig. 2a. For example, CY007458 shows better stabilization ability than CY007476, according to Fig. S4, but their translocation activities seem to be reversed.

6) Enhancement of the protein-protein interaction between 14-3-3beta and CFTR by CY007424 in cells needs to be validated. Coimmunoprecipitation experiment would be a straightforward experimental method. The results will also help authors provide an insight into the cell permeability of compound.

Reviewer #2 (Remarks to the Author):

The paper of Stevers et al focuses on CFTR, the chloride channel known for a large number of pathological mutations underlying cystic fibrosis. The majority of the mutations are related to trafficking defects. Indeed, the drugs that have reached the market so far are mainly chaperons, molecules that improve protein folding and forward trafficking of the mutated channels. The Authors propose that a second class of molecules should be investigated as potential drugs, those that stabilize protein-protein interactions (PPI). Successful examples outside the CFTR are rapamycin and lenalomide.

It is known already that 14-3-3 proteins, a class of regulators of Ser/Thr- phosphorylated proteins, bind to the disordered regulatory domain (R domain) of CFTR and promote forward trafficking of wt and F508-del mutant, the most common mutation of CFTR found in Cystic Fibrosis patients. The R domain is found in between the two cytosolic nucleotide binding domains (NBD1 and 2) and is phosphorylated by PKA during cAMP-regulated channel activation.

The Authors have previously shown that the natural compound fusicoccin (FC-A), a known stabilizer of 14-3-3 complex with their target peptides, stabilizes also the complex formed by 14-3-3 and CFTR, promoting its forward trafficking to the plasma membrane (PM).

Given the structural complexity of FC which constitutes a challenge for medicinal chemistry, the Authors set up a search for novel compounds. They screen a commercial macrocycle library (>5700 compounds) with a fluorescent polarization (FP) assay for binding of a di-phosphorylated peptide derived from CFTR (CFTR pS753pS768) to 14-3-3-beta proteins (14-3-3 have several isoforms and beta is highly expressed in lung epithelium). The screening which occurred in two steps, the initial library of >5700 compounds and a second validation library of 480 compounds, resulted in the identification of 15 compounds (7+8) that reduced the apparent K_d for the CFTR peptide binding to 14-3-3. They don't say of how much exactly this reduction was and the reader should calculate roughly by eye the K_d from the binding curves (Fig. S2, S3, S4).

For one of these compounds, CY007424 they obtained the crystal structure of the complex formed with 14-3-3 and the CFTR peptide. Binding of CY007424 occurs in a different mode than that previously reported for FC. Accordingly, the two molecules have an additive effect increasing the apparent affinity by three orders of magnitude, from μM to nM , confirming that they occupy different binding sites.

Twelve compounds validated from the initial screening are then tested for trafficking of the mutant CFTR channel F508-del in baby hamster kidney (BHK) cells. Surface expression is tested by immunocytochemistry using the HA-tag-CFTR that is exposed towards the extracellular side of the membrane. Three compounds (CY007424, 7288, 7476, 7491) increased membrane surface expression of the mutant channel by about 40%. No control is added to this experiment. In another assay, based on membrane potential readout, the only effective compound becomes CY007424, while the others are not. Why? Do they prevent channel activity? This is not further investigated, nor commented.

The effect on membrane potential of 7424 as well as of the reference compound VX-809 is blocked by channel inhibitor INH172. The two effects, of CY007424 and VX-809, are more than additive. Synergistic is also their effect on ion transport, which was further evaluated across epithelial tissue with the Ussing chamber. In this assay, CY007424 indeed shows NO EFFECT by itself.

I have a main criticism on this study and several minor criticisms listed below.

The general criticism is that 14-3-3 are known to interact with a high number of proteins, among which also membrane proteins such as channels, pump and transporters. Their binding sites are quite broad as well as their binding modalities, and for this reason their list of clients is quite long. Moreover, FC is known to stabilize several of these interactions. Therefore, any new molecule that stabilizes the complex of 14-3-3 with a peptide, in this case form CFTR, must be validated with other known protein interactors of 14-3-3. The risk for aspecific effects is very high. This manuscript doesn't mention nor explore such a possibility that in my opinion, this is an important aspect of this study. I consider therefore the claim for CFTR improved localization and activity farfetched. In addition, CY007424 does not increase activity directly, although it increases the effect of VX-809 by unknown mechanism. The indication on changes in activity presented here are indirectly pointing at an increased conductance of the membranes, not specifically-related to CFTR activity. Claims on activity should be validated by electrophysiology on CFTR.

Minor points

Fluorescent Polarization: is it known at which concentration the 14-3-3 proteins dimerize? Is this affecting the measurement? Is this considered in the interpretation of the experiments of Fig S1-S3-S4?

Figure S3B: why the control no 14-3-3beta, shows binding curves for some of the compounds? Do they bind to the CFTR peptide even in the absence of 14-3-3? Is not the CFTR peptide unfolded?

Figure S1-S3-S4: It would be important to know the estimated Kd values of these compounds and which model is used to fit the data. There are unexpected, at least to me, phenomena going on, as many of the curves do not saturate but become linear (CY007493, 7502) at high protein concentration. CY007424 is even showing a negative slope at high protein concentrations. What does it mean that at high protein concentration binding decreases? These aspects should be mentioned and discussed in the text.

Crystal structure: The crystal shows that two peptides occupy the two grooves of the 14-3-3 dimer and each one is stabilized by a molecule of CY007424.

Is this configuration physiological, given that CFTR, which assembles from a single monomer, has only one peptide that can bind 14-3-3.

Apparent affinity of the new compounds and their additivity with FC:

Page 5: it is mentioned that CY007424 has a Kd of 17 μ M. Still the Kd of the other 7 compounds, isolated from the library, is unknown. Also, it would be interesting to know if they have also been tested for an additive effect with FC.

Trafficking to PM: it would be important to show images of the experiment and not only mean %.

Surface expression: Fig 2a- the effect of FC on this experiment, should be shown for comparison

Reviewer #3 (Remarks to the Author):

The authors report the identification and biochemical, structural, and cell biological characterization of a new macrocycle 'molecular glue' that enhances the interaction of CFTR with 14-3-3beta,

favoring biogenesis. This work builds on earlier studies that have shown (1) the potentially stabilizing interaction of CFTR with 14-3-3, (2) the ability of the natural compound fusicoccin A to stabilize this interaction, and (3) the structural details of the three-way interaction.

This work extends the previous studies in several distinct ways:

1) While FC-A is synthetically challenging, the scaffolds developed here are amenable to systematic synthetic modification, permitting multiple rounds of target refinement, and ultimately the identification of a bona fide CFTR modulator: CY007424. Together with crystallographic data that reveal the stereochemistry of CY007424's interaction with the CFTR and 14-3-3 peptide, this approach opens the door to additional structure-based enhancements in the future.

2) CY007424 has a much stronger effect stabilizing the affinity of CFTR for 14-3-3 than FC-A: >300-fold, vs. ~5-fold. Because FC-A and CY007424 engage the CFTR-14-3-3 complex at separate binding sites, they can be deployed in combination, which further potentiates their individual effects on the affinity of CFTR for the 14-3-3 reporter peptide (~1000-fold), although it is not clear from subsequent studies whether this increase is physiologically significant.

3) The authors show that CY007424 can act additively with VX-809 (Lumacaftor) to further increase rescue of F508del in both BHK and CFBE cell lines. This suggests that combination therapies may be possible.

The work has been performed and presented to a high technical standard. Experiments and results are clearly described, and in general, the systems deployed are appropriate.

However, most of the essential findings represent quantitative, rather than qualitative, advances over results already described with respect to FC-A. The earlier work already demonstrates that a molecular glue can stabilize the target interaction, reveals a stereochemical MOA (albeit at a different site), and shows CFTR rescue in BHK cells. The new compounds have a stronger effect on the affinity of the CFTR interaction with 14-3-3, are synthetically tractable, and bind at a different site. The rescue experiments were extended in this work to include CFBE cells. While welcome advances, none of these points reveals unexpected features of the interactions or their cell-biological implications.

The most significant new finding is the evidence of additivity with VX-809. However, the make-or-break experiments for this approach would test a combination therapy benchmarked against the

current standard of clinical care for most F508del CF patients (Trikafta, not VX-809) and would perform these tests in primary cells. Without such evidence, the likelihood of pharmacological impact remains unclear. In particular, it is possible that tezacaftor or elexacaftor – the correctors included in Trikafta – may themselves strengthen the 14-3-3 interaction, which might well limit the opportunity for additive benefit in combination with a CY007424 derivative. Finally, the EC50 of CY007424, although not explicitly calculated, is likely to be well above 10 microM. Achieving pharmacologically relevant concentrations for such large compounds may be challenging.

Minor queries/typos:

Methods: It is unclear whether the test set of reflections was selected randomly or in thin shells to avoid the influence of non-crystallographic symmetry coupling to the working set. If the NCS nearly creates a pseudosymmetry, the risk is more pronounced. In any case, the selection criterion should be specified.

Which 2Fo-Fc map is shown in Fig. 1b? Is it the electron density that was observed before a model was included, or is it the density seen at the conclusion of refinement? The same question applies to Supplemental Fig. 1c. At least one of the two should show the “unbiased” density observed prior to inclusion of the compound in the model. In addition, each figure should specify the nature of the electron density map shown.

Line 57 (26-30°C.) – no period?

Line 80 1.78Å. Table S1 shows a cutoff of 1.76Å.

Table S1

- “P” in space group should be italics.
- In orthorhombic space groups, it’s typical to assign the unique symmetry axis (here: 2) as the c-axis.
- the Rmeas line is missing a closing parenthesis for the high-resolution shell statistic.

Fig. 1d: Based on earlier work, presume the purple sticks are the FC-A structure, but this is not described in the legend.

Line 120 10microM – missing space.

Line 310 “were” added.

Reviewer #1 (Remarks to the Author):

In this manuscript, Stevers et al. describe the discovery of macrocyclic compound that upregulates the interaction of 14-3-3 and a phosphopeptide fragment of CFTR. The crystal structural analysis of 14-3-3 β bound to the phosphopeptide CFTRpS753pS768 and CY0072424 reveals the new mechanism of stabilizing 14-3-3-ligand interaction upon the formation of ternary complex: The compound binds to distinct part of 14-3-3 and the peptide than fusicoccin-A (FC-A) does, resulting in the remarkable stabilizing effect. The results of biological evaluation using BHK cells expressing 3HA-tagged F508del-CFTR show that CY007458 resulted in the highest signal of membrane-recruited CFTR molecules and their channel activity, suggesting that the compound promotes translocation of CFTR to plasma membrane.

The presented results overall are very interesting. Particularly, the structural analysis study based on the X-ray crystallography have clearly revealed the new mechanism of action of the macrocycle for stabilizing 14-3-3 interaction. This results in the significant enhancement of the peptide binding by 3-ordered magnitude compared to peptide alone, which is very impressive. Intriguingly, the synergetic effect of CY0072427 and VX-809 on the efflux of chloride ion suggests that translocation of mutated CFTR is promoted by the enhancement of 14-3-3-binding in the presence of CY0072427, and that proper folding of the resulting translocated CFTR is assisted by VX-809, leading to better biological outcome compared to VX-0-809 alone. This result suggests the potential clinical application of combination therapy of 14-3-3-enhancer and the conventional CFTR-correctors for cystic fibrosis. Positive modulation of protein-protein interactions still remains challenging task. In my opinion, this paper is sufficiently interest to be accepted after some points have been resolved or corrected. → We thank the reviewer for the time taken to review our manuscript and we are of course happy to read the positive overall evaluation. In the following we will address the individual points the reviewer raised for further improvement.

1) In Fig. 1: Please add the sequential information of the fluorescently labeled peptide used in the screening (S.I.) and the FP assay (Fig. 1e). According to the description "CFTRpS753pS768", the peptide seems to be doubly phosphorylated, whereas the peptide shown in the crystal structure (green sticks) is mono phosphorylated. Please clarify the difference.

→ The sequence information has been added in Figure 3 (formerly Figure 1) (Fig. 3a). In the structure, CY007424 binds only to the CFTRpS753 site. The structure was nonetheless solved with the doubly-phosphorylated peptide CFTRpS753pS768 and the second site – CFTRpS768 – is bound in the other monomer, which doesn't show binding of CY007424 (additionally testifying to a site-selective mode of action of CY007424) (See Figure 3d for the structure with 14-3-3 bound to the two different CFTR phosphosites). The binding stoichiometry is one doubly-phosphorylated CFTRpS753pS768 peptide to one 14-3-3 dimer. The residues between the two phosphorylation sites are not visible in the electron density (probably due to high mobility) and have therefore not been built into the model.

2) In Fig. S4: Please add the apparent K_d values of each compound. In addition, I wonder why CY007424 presents a bell-shaped curve. This may suggest that the compound may form assembly at the low concentration of 14-3-3 and off from 1:1 binding. Please explain.

→ The values for the apparent K_D s have been added to Fig. S1 (former Fig. S4). The bell-shaped curve is probably due to a low-affinity binding event of CY007424 to 14-3-3 α po which at 14-3-3 concentrations that exceed the compound concentration (100 μ M) leaves less compound available

for stabilization of the peptide binding to 14-3-3. We have added a sentence reflecting this notion to the manuscript (page 9, first paragraph).

3) Does the mutation of F508 affect 14-3-3-binding of CFTR? Would it be possible that CY007424 is selective for mutated CFTR over normal CFTR?

→ It is rather unlikely that the CY007424 is specific for mutated CFTR since the mutations are not part of CFTR that contributes to the binding pocket (CFTRpS753).

4) It might be interesting to see if CY007424 is selective for the CFTR peptide. This is likely the case as the crystal structure reveals the tyrosine residue of the compound makes significant hydrophobic contact with the peptide. I suggest that the authors test several peptides for binding to 14-3-3 in the presence of CY007424.

→ We thank the reviewer with this suggestion. We tested eight 14-3-3 partner protein peptides (BADpS74, Foxo3pS253, Foxo4pT32, IRS1pS374, Mdm2pS166, p53pT387, RND3pS210, and RaptorpS722) binding to 14-3-3 in the presence of 100µM CY007424 and saw only with one of them (BADpS74) a stabilization effect. The results are depicted in an additional figure in the SI (Fig. S2 and textual explanations were added to the main manuscript).

5) In Fig. 2a and b: Please discuss the structure-activity-relationship for the biological activities.

There is inconsistency between the K_d values and the cell-based activities presented in Fig. 2a. For example, CY007458 shows better stabilization ability than CY007476, according to Fig. S4, but their translocation activities seem to be reversed.

→ Since the biochemical stabilization activity shown in Fig. S1 has been determined in a fluorescence polarisation assay using synthetic peptides and the plasma membrane localization was determined using HA-tagged CFTR in a cellular setting, it is not necessarily unexpected that the rank order activity is not in all cases the same. In particular, different cellular availability and stability might be different across the compound collection. Importantly, the results with CY007424 are consistent and this is the central compound of which we can say for sure that we know what the molecular mode of action is, since we have the crystal structure of this compound bound to 14-3-3 and CFTR.

6) Enhancement of the protein-protein interaction between 14-3-3beta and CFTR by CY007424 in cells needs to be validated. Coimmunoprecipitation experiment would be a straightforward experimental method. The results will also help authors provide an insight into the cell permeability of compound.

→ We would like to thank the reviewer for this suggestion. We have performed reciprocal pull-downs with both CFTR and 14-3-3 in the absence and presence of CY007424. In the presence of the compound, a clear increase in the pull-downed protein amounts could be observed, indicating a PPI stabilization between CFTR and 14-3-3 in a cellular context using full-length versions of both proteins. These data were added as figure to former Figure 1 (now Figure 3) and a sentence describing these results was added to the main manuscript.

Reviewer #2 (Remarks to the Author):

The paper of Stevers et al focuses on CFTR, the chloride channel known for a large number of

pathological mutations underlying cystic fibrosis. The majority of the mutations are related to trafficking defects. Indeed, the drugs that have reached the market so far are mainly chaperons, molecules that improve protein folding and forward trafficking of the mutated channels. The Authors propose that a second class of molecules should be investigated as potential drugs, those that stabilize protein-protein interactions (PPI). Successful examples outside the CFTR are rapamycin and lenalomide.

It is known already that 14-3-3 proteins, a class of regulators of Ser/Thr- phosphorylated proteins, bind to the disordered regulatory domain (R domain) of CFTR and promote forward trafficking of wt and F508-del mutant, the most common mutation of CFTR found in Cystic Fibrosis patients. The R domain is found in between the two cytosolic nucleotide binding domains (NBD1 and 2) and is phosphorylated by PKA during cAMP-regulated channel activation.

The Authors have previously shown that the natural compound fusicoccin (FC-A), a known stabilizer of 14-3-3 complex with their target peptides, stabilizes also the complex formed by 14-3-3 and CFTR, promoting its forward trafficking to the plasma membrane (PM).

Given the structural complexity of FC which constitutes a challenge for medicinal chemistry, the Authors set up a search for novel compounds. They screen a commercial macrocycle library (>5700 compounds) with a fluorescent polarization (FP) assay for binding of a di-phosphorylated peptide derived from CFTR (CFTR pS753pS768) to 14-3-3-beta proteins (14-3-3 have several isoforms and beta is highly expressed in lung epithelium). The screening which occurred in two steps, the initial library of >5700 compounds and a second validation library of 480 compounds, resulted in the identification of 15 compounds (7+8) that reduced the apparent Kd for the CFTR peptide binding to 14-3-3. They don't say of how much exactly this reduction was and the reader should calculate roughly by eye the Kd from the binding curves (Fig. S2, S3, S4).

→ We thank the reviewer for the time taken to review our manuscript and we are of course happy to read the positive overall evaluation. Regarding this specific point, we have now added the EC50 values to the curves in figures 2 (formerly Fig.S2) and the apparent Kds to Figure S1 (formerly Figure S3).

For one of these compounds, CY007424 they obtained the crystal structure of the complex formed with 14-3-3 and the CFTR peptide. Binding of CY007424 occurs in a different mode than that previously reported for FC. Accordingly, the two molecules have an additive effect increasing the apparent affinity by three orders of magnitude, from μM to nM , confirming that they occupy different binding sites.

Twelve compounds validated from the initial screening are then tested for trafficking of the mutant CFTR channel F508-del in baby hamster kidney (BHK) cells. Surface expression is tested by immunocytochemistry using the HA-tag-CFTR that is exposed towards the extracellular side of the membrane. Three compounds (CY007424, 7288, 7476, 7491) increased membrane surface expression of the mutant channel by about 40%. No control is added to this experiment.

→ We thank the reviewer for this useful comment and added the DMSO control in Fig. 5 (formerly Figure 2) to which all other data had been normalized to.

In another assay, based on membrane potential readout, the only effective compound becomes CY007424, while the others are not. Why? Do they prevent channel activity? This is not further investigated, nor commented.

→ It is indeed an interesting observation that compounds that facilitate an increase of CFTR plasma membrane localization (like -476 and -491, Figure 5a) don't mediate an equally strong increase in ion transport. We don't have a definite explanation for this and considered that this might be due to

differences in binding mode and affinity (of which we can only be certain in the case of CY007424) that are relayed in a differentiated manner to the different functional outputs, or other, possibly non-target effects of these compounds. (A weaker potency of a compound might for example be enough to enhance membrane transport, while not being effective enough for enhancing the actual functional activity of the CFTR). With these tool compounds now in hand, we envision to embark on such detailed follow-up studies.

What we think is important is that both compounds we have detailed information about the mode-of-binding and stabilization – CY007424 and FC-A – show a consistent behaviour between the localization and the ion-transport assay, as do the control compounds VX-809 and trikafta.

The effect on membrane potential of 7424 as well as of the reference compound VX-809 is blocked by channel inhibitor INH172. The two effects, of CY007424 and VX-809, are more than additive. Synergistic is also their effect on ion transport, which was further evaluated across epithelial tissue with the Ussing chamber. In this assay, CY007424 indeed shows NO EFFECT by itself.

I have a main criticism on this study and several minor criticisms listed below.

The general criticism is that 14-3-3 are known to interact with a high number of proteins, among which also membrane proteins such as channels, pump and transporters. Their binding sites are quite broad as well as their binding modalities, and for this reason their list of clients is quite long. Moreover, FC is known to stabilize several of these interactions. Therefore, any new molecule that stabilizes the complex of 14-3-3 with a peptide, in this case form CFTR, must be validated with other known protein interactors of 14-3-3. The risk for aspecific effects is very high. This manuscript doesn't mention nor explore such a possibility that in my opinion, this is an important aspect of this study.

→ The reviewer is right: specificity of any 14-3-3 PPI-stabilizing compound needs ultimately tested in a more broad panel of 14-3-3/partner protein (peptide) combinations and eventually against the background of the entire 14-3-3 interactome by proteomics studies. Since we feel that the latter is beyond the scope of this manuscript, we tested a number of different 14-3-3 partner protein peptides (BADpS74, Foxo3pS253, Foxo4pT32, IRS1pS374, Mdm2pS166, p53pT387, RND3pS210, and RaptorpS722) binding to 14-3-3 in the presence of 100µM CY007424 and saw only with one of them (BADpS74) a stabilization effect. The results are depicted in an additional figure in the SI (Fig. S2) and shortly discussed in the main text.

I consider therefore the claim for CFTR improved localization and activity farfetched. In addition, CY007424 does not increase activity directly, although it increases the effect of VX-809 by unknown mechanism. The indication on changes in activity presented here are indirectly pointing at an increased conductance of the membranes, not specifically-related to CFTR activity. Claims on activity should be validated by electrophysiology on CFTR.

→ In accordance with these concerns, we have toned down our claims taking into considerations the fact that we cannot know for sure that the observed effects ultimately are due to stabilization of the CFTR/14-3-3 complex alone. As for the electrophysiology: we agree that electrophysiology experiments (patch-clamp) would be a valuable extension of our work shown here, however this expertise is not available in either of our three groups. We have however added new coimmunoprecipitation data that substantiate the proof for the enhancement of the physical interaction between 14-3-3 and CFTR (Fig. 3g).

Minor points

Fluorescent Polarization: is it known at which concentration the 14-3-3 proteins dimerize? Is this affecting the measurement? Is this considered in the interpretation of the experiments of Fig S1-S3-S4?

→ Wang et al. (PNAS 2006 103 (46) 17237-17242) showed that the human 14-3-3 isoforms almost exclusively exist as dimers in solution and nothing we have seen in our own work on 14-3-3 over the past 2 decades contradicts that. Therefore, we think that the measurements for this manuscript were not affected by any dissociation events of the 14-3-3 dimers.

Figure S3B: why the control no 14-3-3beta, shows binding curves for some of the compounds? Do they bind to the CFTR peptide even in the absence of 14-3-3? Is not the CFTR peptide unfolded?

→ The compound-concentration curves shown in Fig. 2b (formerly FigS3B) reflect most probably an aggregation effect of the compound and the fluorescently labelled peptide. It is a known phenomenon that some molecules can show and that leads on a regular basis to false-positive hits in screening of libraries. It is a straightforward control experiment to perform the same compound titration in the absence of 14-3-3 to identify such compounds and to exclude them from further studies; which is what we did for these reasons.

Figure S1-S3-S4: It would be important to know the estimated Kd values of these compounds and which model is used to fit the data.

→ We agree with the reviewer and added both the fitted apparent Kd values and included the model in the methods section.

There are unexpected, at least to me, phenomena going on, as many of the curves do not saturate but become linear (CY007493, 7502) at high protein concentration.

→ We thank the reviewer for sharing this observation. There are indeed some peculiarities of some of the binding curves (apparent biphasic) that could be the result of partial degradation of the di-phosphorylated CFTRpS753pS768 peptide. We therefore performed these measurements with a fresh CFTRpS753pS768 peptide batch again. In these measurements, the above described phenomena are not observed anymore (see the new figure S1 in the SI).

CY007424 is even showing a negative slope at high protein concentrations. What does it mean that at high protein concentration binding decreases? These aspects should be mentioned and discussed in the text.

→ The initial indication for a bell-shaped curve (at the far right of the curve) is probably due to a low-affinity binding event of CY007424 to 14-3-3apo which at 14-3-3 concentrations that exceed the compound concentration (100 μ M) leaves less compound available for stabilization of the peptide binding to 14-3-3. We have added a sentence reflecting this notion to the manuscript.

Crystal structure: The crystal shows that two peptides occupy the two grooves of the 14-3-3 dimer and each one is stabilized by a molecule of CY007424.

Is this configuration physiological, given that CFTR, which assembles from a single monomer, has only one peptide that can bind 14-3-3.

→ The two binding sites visible in the electron density of the crystal structure are from the same peptide, which harbors two phosphorylation sites (CFTRpS753pS768). This constellation is also

possible under physiological conditions where the intrinsically disordered R domain of a CFTR monomer displays also both phosphorylation sites. Furthermore, the peptide we used in this and the PNAS crystallographic study of 2016 comprises the native sequence between the two phosphorylation sites and thus very likely reflects the physiological situation in the full-length protein.

Apparent affinity of the new compounds and their additivity with FC:

Page 5: it is mentioned that CY007424 has a K_d of 17 μ M. Still the K_d of the other 7 compounds, isolated from the library, is unknown. Also, it would be interesting to know if they have also been tested for an additive effect with FC.

→ The K_d described on page 5 and shown in Figure 1f is the apparent affinity of the CFTR peptide with which it binds to 14-3-3. This apparent affinity is increased in the presence of a PPI-stabilizing compound, in the case of FC-A by a factor of 4.5 and in the case of CY007424 by more than 300-fold. This shows how much more potent CY007424 is compared to FC-A. Furthermore, combining FC-A and CY007424 results in an increase of apparent affinity by a factor of 809. We have done these measurements now with a number of the other macrocycles alone and in combination with FC-A. Whereas some of these show stabilization of the 14-3-3/CFTRpS753pS768 complex (Fig. S1), none of these is as active as CY007424.

Trafficking to PM: it would be important to show images of the experiment and not only mean %.

→ This is an assay that is done in a 96-well plate and the measurement takes place in a plate reader and not via microscopy.

Surface expression: Fig 2a- the effect of FC on this experiment, should be shown for comparison

→ FC-A has been measured and the data added to Fig. 5a-c (formerly Fig. 2).

Reviewer #3 (Remarks to the Author):

The authors report the identification and biochemical, structural, and cell biological characterization of a new macrocycle ‘molecular glue’ that enhances the interaction of CFTR with 14-3-3 β , favoring biogenesis. This work builds on earlier studies that have shown (1) the potentially stabilizing interaction of CFTR with 14-3-3, (2) the ability of the natural compound fusicoccin A to stabilize this interaction, and (3) the structural details of the three-way interaction.

This work extends the previous studies in several distinct ways:

1) While FC-A is synthetically challenging, the scaffolds developed here are amenable to systematic synthetic modification, permitting multiple rounds of target refinement, and ultimately the identification of a bona fide CFTR modulator: CY007424. Together with crystallographic data that reveal the stereochemistry of CY007424’s interaction with the CFTR and 14-3-3 peptide, this approach opens the door to additional structure-based enhancements in the future.

2) CY007424 has a much stronger effect stabilizing the affinity of CFTR for 14-3-3 than FC-A: >300-fold, vs. ~5-fold. Because FC-A and CY007424 engage the CFTR-14-3-3 complex at separate binding sites, they can be deployed in combination, which further potentiates their individual effects on the affinity of CFTR for the 14-3-3 reporter peptide (~1000-fold), although it is not clear from subsequent studies whether this increase is physiologically significant.

3) The authors show that CY007424 can act additively with VX-809 (Lumacaftor) to further increase rescue of F508del in both BHK and CFBE cell lines. This suggests that combination therapies may be possible.

The work has been performed and presented to a high technical standard. Experiments and results are clearly described, and in general, the systems deployed are appropriate.

→ We thank the reviewer for the time taken to review our manuscript and we are of course happy to read the positive overall evaluation.

However, most of the essential findings represent quantitative, rather than qualitative, advances over results already described with respect to FC-A. The earlier work already demonstrates that a molecular glue can stabilize the target interaction, reveals a stereochemical MOA (albeit at a different site), and shows CFTR rescue in BHK cells. The new compounds have a stronger effect on the affinity of the CFTR interaction with 14-3-3, are synthetically tractable, and bind at a different site. The rescue experiments were extended in this work to include CFBE cells. While welcome advances, none of these points reveals unexpected features of the interactions or their cell-biological implications.

→ We would reply to this assessment with the notion that this is the first time, a synthetically accessible stabilizer of a 14-3-3 protein-protein interaction has been shown to be cellularly active. Furthermore, this molecule is – as the reviewer appreciates, too, - much more active than the natural product FC-A. Finally, the binding site of the macrocycle has not been targeted with any previously reported chemistry. Given the relatively young field of targeted (and not retrospective) protein-protein stabilization and especially the class of 14-3-3 proteins as potential drug targets, we are convinced that the findings reported in this manuscript will make a significant impact in the drug discovery and cystic fibrosis field and certainly among the 14-3-3 community.

The most significant new finding is the evidence of additivity with VX-809. However, the make-or-break experiments for this approach would test a combination therapy benchmarked against the current standard of clinical care for most F508del CF patients (Trikafta, not VX-809) and would perform these tests in primary cells.

→ We have now also tested the combination of CY007424 and Trikafta in both BHK and CFBE cells. Trikafta and CY007424 showed a strong additive (and possibly synergistic) effect in the membrane ion transport assay in BHK cells, while the combination was less potent than CY007424 combined with VX-809 in CFBE cells. Given the lack of knowledge on the molecular mode of action of these existing CFTR drugs, we logically have to refrain from speculations on the different molecular interplays of CY007424 with these drugs.

Without such evidence, the likelihood of pharmacological impact remains unclear. In particular, it is possible that tezacaftor or elexacaftor – the correctors included in Trikafta – may themselves strengthen the 14-3-3 interaction, which might well limit the opportunity for additive benefit in combination with a CY007424 derivative.

→ This scenario might be an explanation for the missing additive effect of CY007424 and Trikafta in the Ussing chamber experiment. But, given the absence of knowledge of MOA of tezacaftor or elexacaftor, we would like to refrain from “over-interpreting” these results. Considering the prospect of using CY007424 in a therapeutic setting: this is the first report of a macrocycle stabilizing the 14-3-3/CFTR interaction and with CY007424 we are definitely not yet at a development candidate stage. What combination with an existing CF drug would be the best – or maybe rather a mono-therapy – remains to be determined in further studies.

Finally, the EC50 of CY007424, although not explicitly calculated, is likely to be well above 10 microM. Achieving pharmacologically relevant concentrations for such large compounds may be challenging.

→ We have calculated the EC50 of CY007424 in the biochemical assay (FP) to be 36µM. This is certainly still a value too high for a successful pharmacological intervention, however CY007424 is a tool compound and we are convinced that the potency of this scaffold can be increased sufficiently in a dedicated medicinal chemistry optimization campaign.

Minor queries/typos:

Methods: It is unclear whether the test set of reflections was selected randomly or in thin shells to avoid the influence of non-crystallographic symmetry coupling to the working set. If the NCS nearly creates a pseudosymmetry, the risk is more pronounced. In any case, the selection criterion should be specified.

→ The test set of reflections was selected randomly. This information was added to the Crystallography part of Materials and Methods (page 17).

Which 2Fo-Fc map is shown in Fig. 1b? Is it the electron density that was observed before a model was included, or is it the density seen at the conclusion of refinement? The same question applies to Supplemental Fig. 1c. At least one of the two should show the “unbiased” density observed prior to inclusion of the compound in the model. In addition, each figure should specify the nature of the electron density map shown.

→ We have added the specific information about the different maps to the caption in Figure 3b and c (formerly Figure 1). Figure 1b now shows an unbiased Fo-Fc map contoured at 2.5σ directly after the successful molecular replacement using the 14-3-3 dimer as search model and a first Refmac run without the peptide and compound present in the model. Figure 2c then shows the final 2Fo-Fc map contoured at 1σ.

Line 57 (26-30°C.) – no period?

→ This has been corrected

Line 80 1.78Å. Table S1 shows a cutoff of 1.76Å.

→ Thank you for the correction, it is indeed 1.76Å.

Table S1

- “P” in space group should be italics.

→ This has been corrected

- In orthorhombic space groups, it’s typical to assign the unique symmetry axis (here: 2) as the c-axis.

→ This is the way it is reported in XDS and the PDB. We therefore would prefer to not change that in the crystallography table.

- the Rmeas line is missing a closing parenthesis for the high-resolution shell statistic.

→ This has been corrected

Fig. 1d: Based on earlier work, presume the purple sticks are the FC-A structure, but this is not described in the legend.

→ That’s right, this information has been added.

Line 120 10microM – missing space.

→ This has been corrected

Line 310 “were” added.

→ This has been corrected

REVIEWER COMMENTS

Reviewer #1 (Remarks to the Author):

Authors have addressed most of the comments raised in the previous version.

Reviewer #2 (Remarks to the Author):

The Authors have done a good job in clearing out all the critical points that I was indicating. I have no more concerns on the ms.

Reviewer #3 (Remarks to the Author):

Most of the comments from the previous review have been satisfactorily addressed and deleted below. I include below only responses for issues that may require additional modification. My new comments are preceded by ">>" text.

...However, most of the essential findings represent quantitative, rather than qualitative, advances over results already described with respect to FC-A. The earlier work already demonstrates that a molecular glue can stabilize the target interaction, reveals a stereochemical MOA (albeit at a different site), and shows CFTR rescue in BHK cells. The new compounds have a stronger effect on the affinity of the CFTR interaction with 14-3-3, are synthetically tractable, and bind at a different site. The rescue experiments were extended in this work to include CFBE cells. While welcome advances, none of these points reveals unexpected features of the interactions or their cell-biological implications.

☐ We would reply to this assessment with the notion that this is the first time, a synthetically accessible stabilizer of a 14-3-3 protein-protein interaction has been shown to be cellularly active. Furthermore, this molecule is – as the reviewer appreciates, too, - much more active than the natural product FC-A. Finally, the binding site of the macrocycle has not been targeted with any previously reported chemistry. Given the relatively young field of targeted (and not retrospective)

protein-protein stabilization and especially the class of 14-3-3 proteins as potential drug targets, we are convinced that the findings reported in this manuscript will make a significant impact in the drug discovery and cystic fibrosis field and certainly among the 14-3-3 community.

>> I appreciate the value of the results presented and am comfortable deferring to the editorial staff on the suitability of the results for publication in this journal or another venue. I would note that if it turns out that compounds CY007476 and CY007491 do not increase reciprocal co-immunoprecipitation of CFTR and 14-3-3 (see first additional comment below), the impact of the manuscript would increase. In that case, there would be a much stronger mechanistic linkage between in-cell complex formation and channel activity. On the other hand, lack of such data or the alternative observation that complex formation increases without a corresponding increase in channel activity are much less conclusive with regards to the proposed mechanism.

...Minor queries/typos:

Methods: It is unclear whether the test set of reflections was selected randomly or in thin shells to avoid the influence of non-crystallographic symmetry coupling to the working set. If the NCS nearly creates a pseudosymmetry, the risk is more pronounced. In any case, the selection criterion should be specified.

☐ The test set of reflections was selected randomly. This information was added to the Crystallography part of Materials and Methods (page 17).

>> This response is OK if the authors can also confirm that the NCS was not close to creating a rotational (PG) pseudosymmetry that would be likely to “couple” intensities between the test and working sets.

- In orthorhombic space groups, it's typical to assign the unique symmetry axis (here: 2) as the c-axis.

☐ This is the way it is reported in XDS and the PDB. We therefore would prefer to not change that in the crystallography table.

>> Unfortunately, XDS does not automatically select the canonical setting for orthorhombic space groups with only a subset (1 or 2) of 2-fold screw axes. Although XDS presents the canonical option, it doesn't select it automatically; instead, the standard option must be selected manually or applied post-integration by reindexing. Ideally, the PDB record and data table should be corrected to conform to the standard setting, since subsequent analysis programs may not parse the data correctly if they assume canonical settings. If the authors are unwilling to do so, the selection of a non-canonical order of axes should be noted in both the data table (via footnote) and the PDB entry (via REMARK).

>>Additional comments on the new elements added to the revision:

>> The co-immunoprecipitation assay is a very important addition to the manuscript, since it confirms that CY007424 can stabilize the CFTR interaction with 14-3-3 in cells. However, following up in particular on the second comment of Reviewer #2, there is an additional important experiment that is missing to make the fundamental conclusion of this paper: that stabilization of the CFTR complex with 14-3-3 is responsible for the observed effects on CFTR surface localization and channel activity. The FLIPR data show that CY007424, but none of the other macrocycles mediate an effect on channel efficacy, although several macrocycles increase cell-surface CFTR levels. For CY007424 and FC-A, the correlation of surface localization and channel activity is observed. For -491 and -476, it is not. Two in favor, two against the hypothesis. What if -491 and -476 do not, in fact, stabilize the CFTR complex with 14-3-3 in cells, but instead increase cell-surface CFTR levels by a different mechanism? Conversely, what if they do? In the latter case, the value of 14-3-3 stabilization becomes much less clear. Without co-IP data to clarify the correlation between 14-3-3 complex formation and channel activity, at a minimum, it would be important to note that the FLIPR results with -476 and -491 raise questions about the mechanism and/or require the invocation of additional factors to explain.

>> I believe the references in the text to the panels in Fig. 1 (formerly supplementary fig. 1) are out of sync with the figure itself. Panels 1b and 1e do not appear to be referenced at all. Please double-check.

>> On p. 10, it is asserted that CY007424 is “a much more potent compound than FC-A, stabilizing the complex by a factor of more than 300x, whereas FC-A shows an about [sic] 4.5x stabilization...” Strictly speaking, the magnitude of the stabilization is governed by both potency and efficacy, so without additional information, it would be more accurate to say it is a “much more potent and/or efficacious compound.”

>> On p. 12, line 7 "20uM" lacks a space.

Reply to Reviewers

REVIEWER COMMENTS

Reviewer #1 (Remarks to the Author):

Authors have addressed most of the comments raised in the previous version.

That is great to hear, thank you.

Reviewer #2 (Remarks to the Author):

The Authors have done a good job in clearing out all the critical points that I was indicating. I have no more concerns on the ms.

That is great to hear, thank you.

Reviewer #3 (Remarks to the Author):

Most of the comments from the previous review have been satisfactorily addressed and deleted below.

We are happy to read that and will address your remaining points below.

I include below only responses for issues that may require additional modification. My new comments are preceded by ">>" text.

...However, most of the essential findings represent quantitative, rather than qualitative, advances over results already described with respect to FC-A. The earlier work already demonstrates that a molecular glue can stabilize the target interaction, reveals a stereochemical MOA (albeit at a different site), and shows CFTR rescue in BHK cells. The new compounds have a stronger effect on the affinity of the CFTR interaction with 14-3-3, are synthetically tractable, and bind at a different site. The rescue experiments were extended in this work to include CFBE cells. While welcome advances, none of these points reveals unexpected features of the interactions or their cell-biological implications.

\ We would reply to this assessment with the notion that this is the first time, a synthetically accessible stabilizer of a 14-3-3 protein-protein interaction has been shown to be cellularly active. Furthermore, this molecule is – as the reviewer appreciates, too, - much more active than the natural product FC-A. Finally, the binding site of the macrocycle has not been targeted with any previously reported chemistry. Given the relatively young field of targeted (and not retrospective) protein-protein stabilization and especially the class of 14-3-3 proteins as potential drug targets, we are convinced that the findings reported in this manuscript will make a significant impact in the drug discovery and cystic fibrosis field and certainly among the 14-3-3 community.

>> I appreciate the value of the results presented and am comfortable deferring to the editorial staff on the suitability of the results for publication in this journal or another venue. I would note that if it

turns out that compounds CY007476 and CY007491 do not increase reciprocal co-immunoprecipitation of CFTR and 14-3-3 (see first additional comment below), the impact of the manuscript would increase. In that case, there would be a much stronger mechanistic linkage between in-cell complex formation and channel activity. On the other hand, lack of such data or the alternative observation that complex formation increases without a corresponding increase in channel activity are much less conclusive with regards to the proposed mechanism.

Thank you for this insight. We have repeated the co-IP experiment with all three compounds (CY007424, CY007476 and CY007491) and presented the data in figure S4. See below for more comments.

...Minor queries/typos:

Methods: It is unclear whether the test set of reflections was selected randomly or in thin shells to avoid the influence of non-crystallographic symmetry coupling to the working set. If the NCS nearly creates a pseudosymmetry, the risk is more pronounced. In any case, the selection criterion should be specified.

(The test set of reflections was selected randomly. This information was added to the Crystallography part of Materials and Methods (page 17).

>> This response is OK if the authors can also confirm that the NCS was not close to creating a rotational (PG) pseudosymmetry that would be likely to “couple” intensities between the test and working sets.

Yes, we confirm this.

- In orthorhombic space groups, it's typical to assign the unique symmetry axis (here: 2) as the c-axis.

(This is the way it is reported in XDS and the PDB. We therefore would prefer to not change that in the crystallography table.

>> Unfortunately, XDS does not automatically select the canonical setting for orthorhombic space groups with only a subset (1 or 2) of 2-fold screw axes. Although XDS presents the canonical option, it doesn't select it automatically; instead, the standard option must be selected manually or applied post-integration by reindexing. Ideally, the PDB record and data table should be corrected to conform to the standard setting, since subsequent analysis programs may not parse the data correctly if they assume canonical settings. If the authors are unwilling to do so, the selection of a non-canonical order of axes should be noted in both the data table (via footnote) and the PDB entry (via REMARK).

We have chosen to keep the non-canonical order of the axes because changing the axes will not change the scientific results of the structure and it will save the time of redepositing the structure. However, we do acknowledge the reviewers note that this is not the standard way and added a footnote to the data table and a remark to the PDB entry.

>>Additional comments on the new elements added to the revision:

>> The co-immunoprecipitation assay is a very important addition to the manuscript, since it confirms that CY007424 can stabilize the CFTR interaction with 14-3-3 in cells. However, following up in particular on the second comment of Reviewer #2, there is an additional important experiment that is missing to make the fundamental conclusion of this paper: that stabilization of the CFTR complex with 14-3-3 is responsible for the observed effects on CFTR surface localization and channel activity. The FLIPR data show that CY007424, but none of the other macrocycles mediate an effect

on channel efficacy, although several macrocycles increase cell-surface CFTR levels. For CY007424 and FC-A, the correlation of surface localization and channel activity is observed. For -491 and -476, it is not. Two in favor, two against the hypothesis. What if -491 and -476 do not, in fact, stabilize the CFTR complex with 14-3-3 in cells, but instead increase cell-surface CFTR levels by a different mechanism? Conversely, what if they do? In the latter case, the value of 14-3-3 stabilization becomes much less clear. Without co-IP data to clarify the correlation between 14-3-3 complex formation and channel activity, at a minimum, it would be important to note that the FLIPR results with -476 and -491 raise questions about the mechanism and/or require the invocation of additional factors to explain.

We have now performed the same pull-down experiment with -491 and -476 as we did earlier with -424. These two compounds did not show a stabilizing effect of the 14-3-3/CFTR interaction in cells. This could confirm the reviewers hypothesis that -491 and -476 increase cell-surface CFTR levels by a different mechanism than 14-3-3/CFTR stabilization.

>> I believe the references in the text to the panels in Fig. 1 (formerly supplementary fig. 1) are out of sync with the figure itself. Panels 1b and 1e do not appear to be referenced at all. Please double-check.

Thank you for this comment, we have now synced the text with the figures.

>> On p. 10, it is asserted that CY007424 is “a much more potent compound than FC-A, stabilizing the complex by a factor of more than 300x, whereas FC-A shows an about [sic] 4.5x stabilization...” Strictly speaking, the magnitude of the stabilization is governed by both potency and efficacy, so without additional information, it would be more accurate to say it is a “much more potent and/or efficacious compound.”

Thank you, we agree that the correct term here is indeed efficacious. We changed this in the manuscript.

>> On p. 12, line 7 "20uM" lacks a space.

We fixed this.